# Human Papillomavirus Vaccine Hesitancy Highly Evident among Caregivers of Girls Attending South African Private Schools

**DOI:** 10.3390/vaccines10040503

**Published:** 2022-03-24

**Authors:** Tracy Milondzo, Johanna C. Meyer, Carine Dochez, Rosemary J. Burnett

**Affiliations:** 1Department of Public Health, University of Limpopo, Polokwane 0727, South Africa; tmlondzo@gmail.com; 2Division of Public Health Pharmacy and Management, Sefako Makgatho Health Sciences University, Pretoria 0204, South Africa; hannelie.meyer@smu.ac.za; 3South African Vaccination and Immunisation Centre, Sefako Makgatho Health Sciences University, Pretoria 0204, South Africa; 4Network for Education and Support in Immunisation, Department of Family Medicine and Population Health, University of Antwerp, 2000 Antwerp, Belgium; carine.dochez@uantwerpen.be; 5Department of Virology, Sefako Makgatho Health Sciences University, Pretoria 0204, South Africa

**Keywords:** human papillomavirus vaccine, vaccination coverage, South Africa, private-sector schools, reasons, vaccine hesitancy

## Abstract

The viral spread of social media misinformation and disinformation regarding human papillomavirus (HPV) vaccination safety has resulted in widespread vaccine hesitancy and suboptimal HPV vaccination uptake. We previously reported that only 19.4% of age-eligible private school girls in South Africa in 2018 had received ≥1 HPV vaccine dose. Here, we report on reasons given by caregivers for why their daughters were unvaccinated. An online survey targeting caregivers of girls in grades 4–7 attending South African private schools was conducted. Caregivers of unvaccinated girls provided the most important reason for their daughter not being vaccinated by either selecting from a list of coded reasons or providing a free text reason. Free text reasons were analysed, coded and added to the list of coded reasons, which were categorised according to broad themes. Frequency distributions of reasons and categories were calculated. Most reasons were related to vaccine hesitancy (61.4%), followed by lack of access to the vaccine (21.3%) and lack of information (15.7%). HPV vaccination coverage among age-eligible girls can be improved by including private-sector schools in the South African HPV vaccination programme, training healthcare providers to advocate for HPV vaccination and extending HPV vaccination advocacy campaigns to include private-sector educators.

## 1. Introduction

Thanks to the coronavirus disease 2019 (COVID-19) pandemic, there are very few people globally who have never heard about vaccines and vaccination. While the vast majority of populations globally are clamouring to get vaccinated against COVID-19, a significant proportion are vaccine hesitant [1]. Vaccine hesitancy, defined by the World Health Organization (WHO) in 2015 as a “delay in acceptance or refusal of vaccination despite availability of vaccination services” [2], is, however, not a new phenomenon arising from the COVID-19 pandemic. It has been around since Edward Jenner’s first vaccine against smallpox [3], and while the advent of the internet allowed vaccine hesitancy to grow, the advent of online social media has allowed vaccine hesitancy to flourish to the extent that it is recognised as a threat to global public health [4,5].

There is overwhelming evidence of the safety of human papillomavirus (HPV) vaccines [6], which are primarily aimed at preventing cervical cancer. However, the viral spread of misinformation and disinformation on social media regarding the safety of HPV vaccination has resulted in widespread vaccine hesitancy [4]. Consequently, there is suboptimal HPV vaccination uptake in many countries globally, including South Africa [7,8,9], a country which bears a disproportionately high burden of cervical cancer in terms of both incidence and deaths [9,10]. Although two HPV vaccines (bivalent and quadrivalent) have been available in South Africa since 2008 [8], it was only in 2014 that the uptake of the bivalent vaccine by the target population (i.e., girls aged 9–13) became widespread. This was because the South African National Department of Health (NDoH) responded to the high burden of cervical cancer by introducing a free, national public-sector-schools-based bivalent HPV vaccination programme in 2014, targeting girls aged ≥9 years in grade 4 [7]. Although the uptake of the first dose in 2014 was relatively high at 86.6% [7], the WHO and United Nations Children’s Fund (UNICEF) estimates of national HPV immunisation coverage have reported a steady decline of two-dose coverage, from 65% in 2014 to 61% in 2015–2016, and 56% in 2017–2019 [11]. While there is anecdotal evidence that vaccine hesitancy played a role in the decline, data on the extent of the contribution made by vaccine hesitancy are lacking [8].

We previously reported that only 19.4% of age-eligible private-sector school girls, whose caregivers were surveyed in 2018, had received ≥1 dose of HPV vaccine [9]. Free HPV vaccination is not offered in South African private-sector schools, which in part explains this low figure. We identified caregivers’ negative attitude towards HPV vaccines as a strong predictor of their daughters being unvaccinated, with misinformation being the main driver of negative attitudes. At the time of publication, the reasons given by caregivers for not vaccinating their daughters had not yet been analysed. This brief research report provides the results of this analysis, which aimed to determine the frequency distribution of reasons given by caregivers for why their daughters aged ≥9 years in grades 4 to 7 attending private schools in South Africa in 2018, were not vaccinated. These results, taken together with those reported previously [9], will allow the NDoH to implement interventions aimed at increasing public demand for HPV vaccination, thereby increasing HPV vaccination coverage among age-eligible girls in South Africa.

## 2. Methods

### 2.1. Study Population and Study Design

The study population for this sub-study was confined to caregivers who had provided reasons for not vaccinating their daughters when responding to a cross-sectional online survey, which has previously been fully described [9]. In summary, after initially failing to obtain a good response when targeting all caregivers of girls aged ≥9 years in grades 4 to 7 attending private schools in 4 provinces of South Africa (Gauteng, Western Cape, North West and Limpopo) in 2018, the online survey was advertised on Facebook, thereby targeting caregivers from all 9 provinces who were Facebook users [9].

A mixed-methods study design was employed for this sub-study nested within the larger cross-sectional survey. First, qualitative data analysis was used for categorising reasons provided via free text responses, and second, quantitative data analysis was used for analysing the frequency of reasons categories and investigating associations between reasons categories and sociodemographic variables of special interest.

### 2.2. Data Collection

Data collection has previously been fully described [9]. In summary, initially, the school principals of 904 schools in 4 provinces were invited via email to forward an invitation to the relevant caregivers of age-eligible girls. Since the response from principals to this email invitation was very low, the survey was linked to a paid Facebook advert targeting all caregivers in all 9 provinces, placed on the South African Vaccination and Immunisation Centre’s Facebook page (Available online: https://www.facebook.com/savicinfo/). Information on the aim and objectives of the study was given, and caregivers who wished to participate either through Facebook or the email invitation gave informed consent via a statement with an option to accept or decline participation in the study, with a link to the online survey if they consented to participate [9]. This study was approved by the University of Limpopo Turfloop Research Ethics Committee (TREC/289/2017/PG).

### 2.3. Data Collection Tool

The online survey was created using the premium version of SurveyMonkey^®^ (Available online: https://www.surveymonkey.com/ (accessed on 23 March 2022)) and pre-tested as previously described [9]. In addition to the items covering sociodemographics, knowledge, attitudes and practices in the online survey [9], there was an item specifically targeting caregivers whose daughters had not received any dose of the HPV vaccine or who were unsure if their daughters had received any dose of HPV vaccine. These caregivers were asked to select the most important reason for their daughter not being vaccinated from a list of reasons (see Table 1). In addition to this list, at the end there was an “other (please specify)” option, which allowed caregivers to provide free text reasons if their reason did not correspond with any of the reason options in the list.

### 2.4. Data Analysis

The comma-separated (CSV) data converted to Microsoft Excel^®^ (Microsoft Office, Redmond, WA, USA) were downloaded from SurveyMonkey^®^ for further analysis. In addition to the descriptive and inferential data analyses previously described [9], the reasons data required the further three steps as follows:

Analysis of “free text” reasons: A qualitative data analysis approach was used, in which the first author analysed the texts to identify and assign codes to themes evident in the data. These themes and codes were checked by the last author, and where there were disagreements between the two authors, the second author checked and made the final decision.

Categorisation of all reasons codes: Following coding of free text answers, these codes were added to the Microsoft Excel^®^ database, allowing all reasons themes to be categorised according to the predominant broad category, by the last author. These broad categories of themes and category codes were checked by the first author, and where there were disagreements between the two authors, the second and third authors were consulted, with the second author making the final decision. Finally, a new summary (category) field was added to the database, using a formula based on the reasons codes in order to automate the generation of categorical data for broad reasons themes.

Frequency distribution analyses: The Microsoft Excel^®^ data were imported to Epi Info^TM^ version 7.2.4.0 (Centers for Disease Control and Prevention, Atlanta, GA, USA) for descriptive statistical analyses of the categorical data for both coded reasons (Table 1 and Table 2) and reasons categorised into broad themes. Frequency distributions using percentages and 95% confidence intervals (95% CI) were calculated. In addition, the categorised reasons data were stratified by willingness to vaccinate if the HPV vaccine was provided free of charge and at their daughter’s school (respectively, 44.3% and 41.1% of respondents of unvaccinated girls as previously reported [9]).

Measurement of associations: Inferential statistical analysis was used to measure associations between reasons categories and caregivers’ (a) employment status, (b) educational level and (c) medical insurance coverage of their daughters. Associations were measured using odds ratios (ORs), 95% CIs around the ORs and chi-square *p*-values, with statistical significance set at *p* < 0.05.

## 3. Results

Of the 333 caregivers whose daughter was not vaccinated/unsure of vaccination status, 319 answered the question on the reasons. In addition, one caregiver (mother) whose daughter had received one dose of HPV vaccine at a previous public-sector school without parental consent, also answered this question. Thus, 320 caregivers in total answered this question. The majority (87.2% (279/320)) were aged between 30 and 49 years, with 6.9% (22/320) aged <30 years and 5.9% (19/320) aged >49 years. All caregivers had some level of secondary education, while 88.8% (284/320) had some level of tertiary education, with 76.6% (245/320) having qualified at tertiary level and 30.0% (96/320) having obtained a postgraduate degree. Data on income levels were not collected; however, 88.1% (282/320) were employed, 8.1% (26/320) were unemployed but not seeking employment, 3.1% (10/320) were unemployed and seeking employment, and 0.6% (2/320) were retired. Of the daughters of these caregivers, 84.4% (270/320) had medical insurance.

Of caregivers providing reasons, 85% (272/320) selected a reason from the drop-down menu (i.e., provided a coded reason), while 15.0% (48/320) selected “other”, 47 of whom provided one or more reasons in full text. Thus, 47 “other” reasons could be coded, making the denominator for the coded reasons 319 instead of 320.

### 3.1. Free Text Reasons

Respondents who gave more than one free text reason: Of the respondents providing free text reasons, 17.0% (8/47) used the opportunity to give more than one reason. All except one of these reasons corresponded to those available in the drop-down menu (see Table 1). For example, the reason (quoted verbatim) “Based on the statistical models used for the prevalence and incidence of HPV infection and cervical cancer complications of HPV, I find that my daughter is in the lowest possible risk group for contracting HPV. Her statistical risk of incurring vaccine damage is statistically higher for her demographic” was coded as both “Daughter is not at risk” and “V vaccine is not safe”. See Table 2.

Respondents who gave only one reason: Other respondents used the free text opportunity to give a more comprehensive explanation for a reason that was in the drop-down menu, with 59.6% (28/47), e.g., “Unsafe and unnecessary. pap smear picks it up without side effect of being paralysed, brain damaged, have seizures and basically mess your health up for ever from age 12. It is sadistic that people do this to their children” was coded as “Vaccine is not safe”. A further 23.4% (11/47) gave a reason that was not in the drop-down menu.

Creation of new codes: Three new codes that were not in the drop-down menu were created for “Vaccine too expensive”; “Need more information”; and “Own extensive research/many reasons”.

### 3.2. Frequency Distribution of Reasons and Broad Reasons Categories

Table 1 includes the frequency distribution of all the reasons in the drop-down menu that were available for selection by respondents, while Table 2 provides the final frequency distribution of reasons following coding of the “other” free text reasons. In addition, analysis after separation of codes where respondents gave more than one “other” reason resulted in the following: 30.1% (96/319) said the vaccine is not safe; 16.6% (53/319) had not had the chance to vaccinate their daughters yet; 10.0% (32/319) said the vaccine is too new; 5.0% (16/312) said the vaccine is ineffective; 1.9% (6/319) could not afford the vaccine; 1.3% (4/319) had been advised by their healthcare provider not to vaccinate their daughter; and 1.3% (4/319) said there were too many vaccines.

Together with the 3 additional codes created following the free text analysis, there were 14 codes in total. These 14 codes were further categorised into four broad categories as follows:

Reasons related to vaccine hesitancy: This was the most common category, provided by 61.4% (196/319) of respondents, and included: My daughter is too young; My daughter is not at risk; I do not think the vaccine is safe; I do not think that vaccines are effective/provide long-term protection; There are too many vaccinations; Vaccine is too new/no evidence it works/more research is needed; My healthcare provider has advised me against it; Many reasons/my own extensive research. Of the respondents who cited reasons related to vaccine hesitancy, 91.8% (180/196) had some level of tertiary education; 90.3% (177/196) of their daughters had medical insurance; 85.7% (168/196) were employed; and 20.4% (40/196) were willing to have their daughters vaccinated if the vaccine was offered free of charge and at their daughter’s school.

Reasons related to lack of access: This category was the second most common, provided by 21.3% (68/319) of respondents, and included: Have not had the chance to do it yet (no opportunity); The vaccine is too expensive; My healthcare provider has not recommended it. Of the respondents who cited reasons related to lack of access, 95.6% (65/68) were employed; 88.2% (60/68) had some level of tertiary education; 83.8% (57/68) were willing to have their daughters vaccinated if the vaccine was offered free of charge and at their daughter’s school; and 80.9% (55/68) of their daughters had medical insurance. Of those who were not willing to vaccinate if the vaccine was provided at their daughter’s school, 54.6% (6/11) had selected the reason “Healthcare provider did not recommend vaccination” as the reason for not being vaccinated, while 18.2% (2/11) were willing to vaccinate if the vaccine was free, both of whom gave the reason that they had not yet had the opportunity to vaccinate their daughter.

Reasons related to no information: This category, provided by 15.7% (50/319) of respondents, ranked third, and included: We never thought about it; I need more information about the vaccine. Of the respondents who cited reasons related to no information, 88.0% (44/50) were employed; 76% (38/50) had some level of tertiary education; 68.0% (34/50) of their daughters had medical insurance; and 68.0% (34/50) were willing to have their daughters vaccinated if the vaccine was offered free of charge and at their daughter’s school.

Reasons related to reportedly genuine precautions or contraindications: This last category, selected by 1.6% (5/319) of respondents, included just one reason: My daughter has an illness/medical condition that precludes vaccination. Of the respondents who cited this reason, 100% (5/5) were employed and had some level of tertiary education; 60.0% (3/5) of their daughters had medical insurance; 60.0% (3/5) were not willing to have their daughters vaccinated if the vaccine was offered free of charge and at their daughter’s school; 20.0% (1/5) was willing to vaccinate if the vaccine was free; and 20.0% (1/5) was willing to vaccinate if the vaccine was offered at school.

The vast majority (98.4% (314/319)) of respondents provided reasons that could be targeted by interventions to increase public demand and HPV vaccination coverage among age-eligible girls in South Africa. There were statistically significant associations between reporting reasons related to vaccine hesitancy and (a) having some level of tertiary education (OR: 2.2; 95% CI: 1.1–4.4) and (b) daughters having access to medical insurance (OR: 3.1; 95% CI: 1.7–5.9). There were also statistically significant associations between reporting reasons related to a lack of information and (a) having some level of tertiary education (OR: 0.3; 95% CI: 0.1–0.7) and (b) daughters having access to medical insurance (OR: 0.3; 95% CI: 0.2–0.6). No other statistically significant associations were identified.

## 4. Discussion and Conclusions

We had previously reported that caregivers’ negative attitudes towards the HPV vaccine were mainly driven by misinformation, resulting in only 19.4% of age-eligible girls attending private-sector South African schools having received ≥1 dose of HPV vaccine [9]. That report is supported by our current finding that 61.4% of respondents whose daughters were unvaccinated cited reasons related to vaccine hesitancy. The major reason was about safety concerns, which was also the major concern when the national public-sector HPV vaccination programme was introduced [7,8]. As previously pointed out, caregivers of private-school girls in South Africa do not receive HPV vaccination information through school authorities, so those who do seek this information may turn to the internet, where they are highly likely to find misinformation and disinformation, leading to vaccine hesitancy [9].

Vaccine hesitancy is a complex human condition, determined by many factors, and ranging from acceptance, even when in doubt about safety and/or effectiveness, to refusal because of these doubts [2]. Our finding that 20.4% of vaccine-hesitant respondents were willing to vaccinate their daughters if the vaccine was provided free of charge and at their daughter’s school, indicates that these caregivers are on the “acceptance even when in doubt” end of the vaccine hesitancy spectrum. Even though none of these respondents cited additional reasons related to lack of access, the prospect of being provided with access appears to be a sufficient “nudge” to move them towards vaccine acceptance. An additional explanation could be that these respondents may regard school authorities as independent and trustworthy decision makers and are thus prepared to take the lead from them. This assumption is supported by a South African study on HPV vaccination uptake by age-eligible public-sector school girls, where advice from teachers was reported to have the most influence over caregivers’ HPV vaccination-related decisions for their daughters [12].

Since the South African national HPV vaccination programme excludes age-eligible private-sector school girls [8,9], caregivers who want their daughters to receive HPV vaccination have to take their daughters to private healthcare providers. Apart from the costs associated with travel and the inconvenience (and perhaps lost wages) of taking time off work, we previously reported that 41% of respondents whose daughters were vaccinated had paid for the full cost of the HPV vaccine. We had also reported that over 40% of respondents whose daughters were unvaccinated were willing to vaccinate them if the HPV vaccine was provided free of charge and at their daughter’s school (respectively, 44.3% and 41.1%) [9]. Given this background, we had expected a greater proportion of reasons for not being vaccinated to be related to lack of access than the 21.3% reported here. Unsurprisingly, the vast majority (83.8%) of respondents reporting reasons related to lack of access were willing to vaccinate their daughters if the vaccine was provided free at their daughter’s school. Most noteworthy is that the major reason for not being willing despite having this free and easy access to HPV vaccination was because their healthcare provider had not recommended HPV vaccination for their daughter. This underscores the important role that healthcare providers play in increasing HPV vaccination uptake [8,9].

One of the major drawbacks of not including private-sector school girls in the national HPV vaccination programme is that their caregivers are not provided with evidence-based HPV vaccination information by the school authorities [8,9]. This explains why 15.7% of respondents whose daughters were not vaccinated reported reasons related to lack of information, mainly that they had “Never thought about it”. Fortunately, the prospect of being given free and easy access to HPV vaccination was a sufficient “nudge” for 68% of these respondents to accept HPV vaccination for their daughters. This, again, suggests that these caregivers have considerable trust in school authorities and are willing to take the lead from them despite lacking knowledge on the interventions they offer.

The only permanent contraindication to the HPV vaccines currently available in South Africa is a known hypersensitivity to any component of the vaccine [13], with allergic reactions or anaphylaxis being a rare adverse event following immunisation with both vaccines [14]. Temporary contraindications include severe febrile illness and pregnancy, although there is no evidence of adverse pregnancy outcomes in women who did not know they were pregnant at the time of vaccination [14]. Since two respondents said they were willing to vaccinate their daughters who had contraindications (one being willing to vaccinate if the vaccine was provided free, while the other was willing if it was provided at their daughter’s school), it seems likely that these were temporary contraindications.

When reporting the results on vaccination coverage for the larger study population in 2021 [9], we had found no statistically significant differences between vaccinated and unvaccinated groups for the sociodemographic variables. For example, of caregivers of vaccinated girls, 86.1% had some level of tertiary education compared to 88.0% of caregivers of unvaccinated girls. In addition, of caregivers of vaccinated girls, 88.6% were employed compared to 88.0% of caregivers of unvaccinated girls. Additionally, of caregivers of vaccinated girls, 82.5% of their daughters had medical insurance compared to 83.8% of caregivers of unvaccinated girls. We, thus, did not see any value in reporting these summary statistics stratified by vaccination status, although they can be calculated from the data we provided in Table 2 of our 2021 report [9]. In contrast, this sub-study, which is confined to caregivers who reported reasons for not vaccinating, found that vaccine-hesitant caregivers have statistically significant higher education levels and more access to medical insurance for their daughters than caregivers who report other reasons for not vaccinating. Furthermore, caregivers who reported a lack of information were statistically significantly less likely to have some tertiary education and access to medical insurance for their daughters than caregivers reporting other reasons. While these sociodemographic factors cannot be targeted by interventions to increase HPV vaccination uptake, these findings add to the body of knowledge on the drivers on HPV vaccine acceptance in South Africa.

### 4.1. Study Limitations

As previously reported [9], this study suffers from sampling bias because it used an online survey that could only be accessed by caregivers with email accounts in the first phase, and Facebook accounts in the second phase. This bias may have resulted in an over-estimation of reasons related to vaccine hesitancy, as caregivers with internet access and those who use social media are more likely to encounter HPV vaccine misinformation and disinformation [4,9]. However, this bias would have been countered by the non-response/volunteer bias that may have been introduced by the refusal of some school principals to participate in the first phase of the study, and the boycott of the second phase called for by anti-vaccination lobbyists, as previously reported [9]. This bias may have resulted in an under-estimation of reasons related to vaccine hesitancy. Additionally, most responses came after placing the Facebook advert, thus the respondents may not all have fitted the inclusion/exclusion criteria, as one cannot verify that Facebook users are in fact who they claim to be. Thus, the results of this study must be treated with caution, as they may not be a true reflection of the frequency distribution of all the reasons why age-eligible girls attending South African private schools were not vaccinated against HPV.

### 4.2. Conclusions and Recommendations

This study confirmed our previous findings that misinformation was the major driver of the very low HPV vaccination uptake by private-sector school girls whose caregivers participated in this study [9]. Social media messaging around the COVID-19 pandemic and subsequent global COVID-19 vaccination campaigns have considerably amplified anti-vaccination lobbying through the dissemination of vaccine misinformation and disinformation. This has the potential to further entrench vaccine hesitancy towards other vaccines, especially the HPV vaccine, since it was previously identified as having the highest rates of vaccine hesitancy [8]. Ultimately, HPV vaccination coverage may drop even lower if timely interventions to increase vaccine confidence are not implemented.

Based on our previous results [9], we had wrongly expected that around 40% of respondents would have cited reasons related to lack of access and that all of those reporting reasons related to lack of access would be willing to vaccinate if provided with free and easy access. Instead, we found that the reasons underlying willingness to vaccinate were more complex, being spread over all categories of reasons. In particular, the finding that 20% of vaccine-hesitant respondents and 68% of respondents who lacked information were willing to vaccinate their daughters if given free and easy access to HPV vaccination services, now provides a more compelling argument for including private-sector schools in the national programme than we had previously put forward [9]. Additionally, the finding that a lack of recommendation from a healthcare provider was the major reason why respondents who lacked access were not willing to vaccinate if given free and easy access, strengthens our argument for training healthcare providers [9]. Finally, while we had called for additional advocacy campaigns to include private-sector educators [9], this study confirms that the trust that caregivers have in school authorities could be leveraged to build public demand for HPV vaccination.

## Figures and Tables

**Table 1 vaccines-10-00503-t001:** Frequency distribution of caregivers’ reasons for being unsure or not wanting to vaccinate before coding of “other” free text reasons (*n* = 320).

Variable	*n* (%)	95% CIs
Vaccine is not safe	76 (23.8)	19.4–28.7
Other *	48 (15.0)	11.5–19.3
Never thought about it	45 (14.1)	10.7–18.3
No chance to vaccinate yet (no opportunity)	44 (13.8)	10.4–18.0
Daughter/ward is too young	42 (13.1)	9.9–17.3
Vaccine is too new	27 (8.4)	5.9–12.0
Vaccine is ineffective/does not provide long-term protection	12 (3.6)	2.2–6.4
Healthcare provider did not recommend vaccination	11 (3.4)	1.9–6.1
Daughter/ward is not at risk of HPV infection and cervical cancer	5 (1.6)	0.7–3.6
Daughter has an illness/medical condition that precludes vaccination	4 (1.3)	0.5–3.2
Healthcare provider advised against HPV vaccine	3 (0.9)	0.3–2.7
There are too many vaccinations	3 (0.9)	0.3–2.7

* Refers to other reasons not listed on the drop-down list.

**Table 2 vaccines-10-00503-t002:** Frequency distribution of caregivers’ reasons for being unsure or not wanting to vaccinate following coding of the “other” free text reasons (*n* = 319).

Variable	*n* (%)	95% CI
Vaccine is not safe	92 (28.8)	24.1–34.0
No chance to vaccinate yet/do want to vaccinate daughter/ward	51 (16.0)	12.4–20.4
Never thought about it	45 (14.1)	10.7–18.4
Daughter/ward too young	42 (13.2)	9.9–17.32
Vaccine is too new/no evidence/need more research evidence about the vaccine	29 (9.1)	6.4–12.8
Vaccine is ineffective	14 (4.4)	2.6–7.2
Healthcare provider did not recommend vaccination	11 (3.5)	1.9–6.1
Vaccine is contraindicated	5 (1.6)	0.7–3.6
Daughter/ward not at risk of HPV infection and cervical cancer	5 (1.6)	0.7–3.6
Need information about the vaccine	5 (1.6)	0.7–3.6
Vaccine is too expensive	4 (1.3)	0.5–3.2
Healthcare provider advised against HPV vaccine	3 (0.9)	0.3–2.7
There are too many vaccines	3 (0.9)	0.3–2.7
Do want to vaccinate daughter/ward but vaccine is too expensive	2 (0.6)	0.2–2.3
Vaccine is not safe and daughter/ward is not at risk of HPV infection and cervical cancer	2 (0.6)	0.2–2.3
Many reasons */own extensive research	2 (0.6)	0.2–2.3
Healthcare provider advised against HPV vaccine and the vaccine is too new and there is no evidence about the vaccine	1 (0.3)	0.1–1.8
Vaccine is ineffective and it is not safe	1 (0.3)	0.1–1.8
Vaccine is ineffective and it is too new and there is no evidence about the vaccine	1 (0.3)	0.1–1.8
Vaccine is not safe and there are too many vaccines and the vaccine is too new and there is no evidence about the vaccine	1 (0.3)	0.1–1.8

* Refers to the drop-down list of reasons for selection by caregivers.

## Data Availability

The datasets generated for this study are available on request to the corresponding author.

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
