# Peer review of "Human Papillomavirus Vaccine Hesitancy Highly Evident among Caregivers of Girls Attending South African Private Schools"

_vaccines, 2022, doi:10.3390/vaccines10040503_

Round 1

Reviewer 1 Report

It is a very meaningful study. The information collected in this study is not only useful for promoting HPV vaccination but may also apply to the promotion of other vaccines.

However, I suggest that the authors carefully discuss the sub-population for which the conclusion applies, such as parents' educational attainment, income level, and access to medical services.

Author Response

Reviewer 1: It is a very meaningful study. The information collected in this study is not only useful for promoting HPV vaccination but may also apply to the promotion of other vaccines. However, I suggest that the authors carefully discuss the sub-population for which the conclusion applies, such as parents' educational attainment, income level, and access to medical services.

Response from the authors: Thanks very much for this suggestion; it has added an important dimension to the report. We have now included additional data analysis details (p4, lines 153-157) and the relevant socio-demographic results for the 320 respondents (p4, lines 163-171). We have further stratified the analysis according to reasons for not vaccinating as follows: respondents giving reasons related to vaccine hesitancy (p5, lines 215-217); respondents giving reasons related to lack of access (p5, lines 223-226); respondents giving reasons related to lack of information (p5, lines 234-235); and respondents giving reasons related to contraindications (p5, lines 240-242). The results of the inferential data analysis have been added (p6, lines 250-256), and these results are contrasted with the previous findings that these sociodemographic variables did not predict vaccination uptake in the coverage study (p7, lines 318-335).    

Reviewer 2 Report

Thanks for the invitation. This is an interesting piece of work, however; a few changes are needed before considering this manuscript for publication.

Title: I suggest to include the name of HPV vaccine in the title of the manuscript

Methods:

Authors need to describe why specific age is considered while selecting the participants i.e. all caregivers of girls aged ≥9 years

Data collection has previously been fully described........ authors are referring to their previous study again and again, however; the manuscript should have clear and compact methodology at the place rather than readers would refer to other paper. Please cover the headings; ethics, study population, study design and site, data collection, study instrument, operational definitions, statistics

There is no detail of the study instrument, I have confused here, in this study a quantitative survey or qualitative analysis? authors are performing thematic analysis; at the same time, they provided the respondents with the list of reasons for not vaccinating their girls. If this is a qualitative analysis then how can quantitative data be obtained via an online survey? For the study questionnaire, there is no information provided that how the tool was developed, validation, reliability, translation, etc...

The authors have conducted a previous study as described in reference no. 9, can the author explain how this study is different from the previous one? What was the need to have another almost similar kind of study, if authors have already published the work? is this study also using the data from a previous study? 

In fact, I would suggest authors avoid too much comparison of results with their own study they have done previously. If this study is entirely different from the previous ones, then only the authors should discuss the previous study in the discussion section.

There is no need for figure 1, rather authors should provide a figure showing the study flow diagram.

authors should discuss the impact of the COVID-19 vaccination drive on the hesitancy and acceptance of the respondents. Since vaccine hesitancy is widely discussed during the pandemic, those who became hesitant of the covid-19 vaccine have also the potential to become hesitant to other vaccines. 

Can the authors provide more details of the participants? Authors should also share the data collection form/interview guide as a supplementary file. 

Please also provide information that the data collection form was advertised by link or service provided prepared in its own domain?

The discussion section is well written....

Author Response

Reviewer 2: Thanks for the invitation. This is an interesting piece of work, however; a few changes are needed before considering this manuscript for publication.

Title: I suggest to include the name of HPV vaccine in the title of the manuscript

Response from the authors: Thanks very much for this suggestion; this has been addressed (p1, lines 2-4).

Methods:

Authors need to describe why specific age is considered while selecting the participants i.e. all caregivers of girls aged ≥9 years

Response from the authors: HPV vaccines are licensed from the age of 9 years, and as discussed in the manuscript, in South Africa, the National Department of Health targets cervical cancer prevention through free provision of the bivalent vaccine to girls aged ≥9 years. These details are provided on p2, lines 52-56.

Data collection has previously been fully described........ authors are referring to their previous study again and again, however; the manuscript should have clear and compact methodology at the place rather than readers would refer to other paper. Please cover the headings; ethics, study population, study design and site, data collection, study instrument, operational definitions, statistics.

Response from the authors: We have now clarified these issues by describing in more detail (a) the specific sub-population (i.e. caregivers who gave reasons why their daughters are not vaccinated) of the parent study (p2, lines 77-79); (b) the mixed-methods study design (p2, lines 84-88); (c) the “reasons” data collection tool (p3, lines 116-126); and (d) statistical methods (p4, lines 153-157). Details on the ethical issues of informed consent (p3, lines 103-107) and ethical clearance (p3, lines 113-114) have been provided.   

There is no detail of the study instrument, I have confused here, in this study a quantitative survey or qualitative analysis? authors are performing thematic analysis; at the same time, they provided the respondents with the list of reasons for not vaccinating their girls. If this is a qualitative analysis then how can quantitative data be obtained via an online survey? For the study questionnaire, there is no information provided that how the tool was developed, validation, reliability, translation, etc...

Response from the authors: In addition to providing clarity on the study design and data analysis (see above response), details of the data collection tool have now been moved to a separate section, and expanded upon (p3, lines 116-126).

The authors have conducted a previous study as described in reference no. 9, can the author explain how this study is different from the previous one? What was the need to have another almost similar kind of study, if authors have already published the work? is this study also using the data from a previous study?

Response from the authors: We have now further clarified that this is a sub-study of the previously published parent study (p2, lines 77-79). This study is confined to the sub-population of caregivers who had provided reasons for their daughters being unvaccinated. These data on reasons were not included in the previous publication.

In fact, I would suggest authors avoid too much comparison of results with their own study they have done previously. If this study is entirely different from the previous ones, then only the authors should discuss the previous study in the discussion section.

Response from the authors: Since this is a sub-study of the larger parent study, with data for the sub-study being collected as part of the parent study, the parent study is cited only when relevant and necessary. The previous report was on knowledge and attitudes of caregivers regarding HPV vaccination, and HPV vaccination coverage of their daughters. This sub-study reports on reasons given by caregivers for not vaccinating their daughters. It is thus not possible to compare results of this sub-study to those of the parent study.

There is no need for figure 1, rather authors should provide a figure showing the study flow diagram.

Response from the authors: Since the results given in figure 1 are not reported elsewhere in the text, we would prefer to retain this figure. Hopefully the clarifications provided in this updated manuscript provide all the relevant details regarding the flow of this sub-study.

authors should discuss the impact of the COVID-19 vaccination drive on the hesitancy and acceptance of the respondents. Since vaccine hesitancy is widely discussed during the pandemic, those who became hesitant of the covid-19 vaccine have also the potential to become hesitant to other vaccines.

Response from the authors: This has been addressed; see p8, lines 357-363.

Can the authors provide more details of the participants? Authors should also share the data collection form/interview guide as a supplementary file.

Response from the authors: Additional details have been added on the study participants (p2, lines 77-79), while Table 1 (p10, lines 435-436) provides the full list of options for selection contained in the data collection tool on which this sub-study is based. 

Please also provide information that the data collection form was advertised by link or service provided prepared in its own domain?

Response from the authors: The details of the Facebook page have been added (p3, lines 102-103)

The discussion section is well written....

Round 2

Reviewer 2 Report

The authors have addressed all the comments, except one. They should consider omitting figure 1, as Pie charts are widely discouraged from presenting the data. Moreover, the figures are only considered necessary if the data is too verbatim to present in the text. In this figure, the authors have just described the frequencies which could be easily presented in the text. If authors really wish to have any figure in the manuscript, then I suggest considering including the Figure on Study flow diagram to summarize how the study was conducted. 

Author Response

The authors have addressed all the comments, except one. They should consider omitting figure 1, as Pie charts are widely discouraged from presenting the data. Moreover, the figures are only considered necessary if the data is too verbatim to present in the text. In this figure, the authors have just described the frequencies which could be easily presented in the text. If authors really wish to have any figure in the manuscript, then I suggest considering including the Figure on Study flow diagram to summarize how the study was conducted.

Response from authors: Thanks very much for this advice; we have now removed Figure 1 and included the results in the text. Please see p5, lines 198-199; lines 208-209; lines 220-221; line 228; and lines 235-236. 
